# Pharmacogenomics of Pediatric Cardiac Arrest: Cisplatin Treatment Worsened by a Ryanodine Receptor 2 Gene Mutation

Angela Maggio [1,†], Sandra Mastroianno [2,†], Giuseppe Di Stolfo [2,*], Stefano Castellana [3,4], Pietro Palumbo [5], Maria Pia Leone [5], Anita Spirito [1], Domenico Rosario Potenza [2], Saverio Ladogana [1], Marco Castori [5], Massimo Carella [3,4], Massimo Villella [2] and Mauro Pellegrino Salvatori [2]

1 Pediatric Oncoematology Unit, Fondazione IRCCS Casa Sollievo della Sofferenza, 71013 San Giovanni Rotondo, Italy; a.maggio@operapadrepio.it (A.M.); a.spirito@operapadrepio.it (A.S.); s.ladogana@operapadrepio.it (S.L.)
2 Cardiology Unit, Fondazione IRCCS Casa Sollievo della Sofferenza, 71013 San Giovanni Rotondo, Italy; sandramastroianno@hotmail.it (S.M.); d.potenza@operapadrepio.it (D.R.P.); m.villella@operapadrepio.it (M.V.); m.salvatori@operapadrepio.it (M.P.S.)
3 Bioinformatics Unit, Fondazione IRCCS Casa Sollievo della Sofferenza, 71013 San Giovanni Rotondo, Italy; s.castellana@operapadrepio.it (S.C.); m.carella@operapadrepio.it (M.C.)
4 Istituto Mendel, 00161 Roma, Italy
5 Genetic Unit, Fondazione IRCCS Casa Sollievo della Sofferenza, 71013 San Giovanni Rotondo, Italy; p.palumbo@operapadrepio.it (P.P.); leonemp87@gmail.com (M.P.L.); m.castori@operapadrepio.it (M.C.)
* Correspondence: giuseppedistolfo@yahoo.it
† These authors contributed equally to this work.

**Abstract:** In thelast few decades, the roles of cardio-oncology and cardiovascular geneticsgained more and more attention in research and daily clinical practice, shaping a new clinical approach and management of patients affected by cancer and cardiovascular disease. Genetic characterization of patients undergoing cancer treatment can support a better cardiovascular risk stratification beyond the typical risk factors, suchas contractile function and QT interval duration, uncovering a possible patient's concealed predisposition to heart failure, life threatening arrhythmias and sudden death. Specifically, an integrated cardiogenetic approach in daily oncological clinical practice can ensure the best patient-centered healthcare model, suggesting, also the adequate cardiac monitoring timing and alternative cancer treatments, reducing drug-related complications. We report the case of a 14-month-old girl affected by neuroblastoma, treated by cisplatin, complicated by cardiac arrest. We described the genetic characterization of a Ryanodine receptor 2 (*RYR2*) gene mutation and subsequent pharmacogenomic approach to better shape the cancer treatment.

**Keywords:** cardiac arrest; cisplatin; ryanodine receptor 2 gene mutation; pharmacogenomics; neuroblastoma

## 1. Background

In the last few decades, the roles of cardio-oncology and cardiovascular geneticsgainedmore and more attention in research and daily practice, shaping a new clinical approach and management of patients affected by cancer and cardiovascular disease. As it is common practice when something new is discovered, scientists and clinicians followsuper specialized pathways to investigate in detail each single brick of the new process, while the final aim is to manage each patient in their own fragmented complexity [1].

In this regard, the genetic characterization of patients undergoing cancer treatment could allow a better cardiovascular risk stratification beyond the typical risk factors, such as contractile function and QT interval duration, uncovering a possible patient's concealed predisposition to heart failure, life threatening arrhythmias and sudden death. An integrated cardiogenetic approach in daily oncological clinical practice will ensure the best

patient-centered healthcare model, suggesting, also the adequate cardiac monitoring timing and alternative cancer treatment, reducing drug-related adverse events [2].

Although the creation of dedicated registries shared between oncologists, cardiologists and geneticists will be the main tool to increase our knowledge about this issue, case reports represent an important causefor reflection, the first step to guarantee a comprehensive knowledge sharing on the worst case scenarios, as "mistakes are the portals of discovery" [3].

Patient predisposition to heart failure and cardiac arrhythmias relies depend on the strong relationship between genetic features and the specific cancer treatment, the latter representing a genetic modifier; early recognition of these complex interactions will improve personalized treatments and will reduce side effect [4].

## 2. Case Report

We describe the case ofa 14-month-old girl affected by congenital sensorineural deafness, carrying a cochlear implant, who presented with an abdominal and mediastinal mass. Lymphnode biopsy allowed the diagnosis of stage IV, N-Myc-non-amplified (Figure 1) neuroblastoma [5,6]. Results of the preliminary resting electrocardiogram and echocardiogram were normal. After complete staging, she was enrolled in the SIOPEN/HR-NBL1 Protocol (Société International d'Oncologie Pédiatrique European Neuroblastoma/High Risk Neuroblastoma Protocol 1) by COJEC scheme (rapid, platinum-containing induction schedule: CBDCA, CDDP, CYC, VCR, VP16) including cisplatin [7]. During the 7th cycle with cisplatin, on the second day from therapy she had irritability and sudden cardiocirculatory arrest with evidence of ventricular fibrillation (VF). After one hour of continuous cardiopulmonary resuscitation associated to repetitive epinephrine iv and cardiac defibrillation, according to the current guidelines [8], she was returned to spontaneous circulation (ROSC). Then the child was transferred to the intensive care unit. First electrocardiogram (ECG) after cardiac arrest showed QTc prolongation (Figure 2), normalized at subsequent controls, likely due to acute metabolic acidosis and hyperkalemia, while magnesium concentrations were normal before cardiac arrest (2.4 mg/dL, normal range 1.6–2.6 mg/dL). Echocardiography showed normal biventricular function, without any evidence of cardiac structural disease.

Nevertheless, in the presence of sensorineural deafness and evidence of long QT after cardiac arrest, in the absence of familial history of sudden death and deafness, we hypothesized the Jervell and Lange-Nielsen syndrome, an autosomal recessive variant of the familial long QT syndrome; consequently, genetic analysis was performed [9].

On the contrary, the next generation sequencing (NGS) panel for bilateral sensorineural deafness was negative for potassium channel mutation, while it showedhomozygosity for a *GJB2* frameshift pathogenic variant c.35delG, p.(Gly12Valfs*12), implicated in a recessive form of congenital sensorineural deafness [10].

Therefore, we performed the NGS panel for suspected channelopathies showing a missense mutation in the*RYR2* gene (*RYR2*: c.1927T > A; p.(Leu643Ile), classified as a variant of uncertain significance (VUS). The *RYR2* gene provides for making the protein ryanodine receptor 2, involved in calcium transport handling within cardiomyocytes cells. The *RYR2* gene is closely related to catecholaminergic polymorphic ventricular tachycardia (CPVT). According to present consensus, after parents screening, the mother resulted to be a carrier of the same mutation. A more detailed maternal clinical history revealed her positivity to palpitations and two syncopal events.

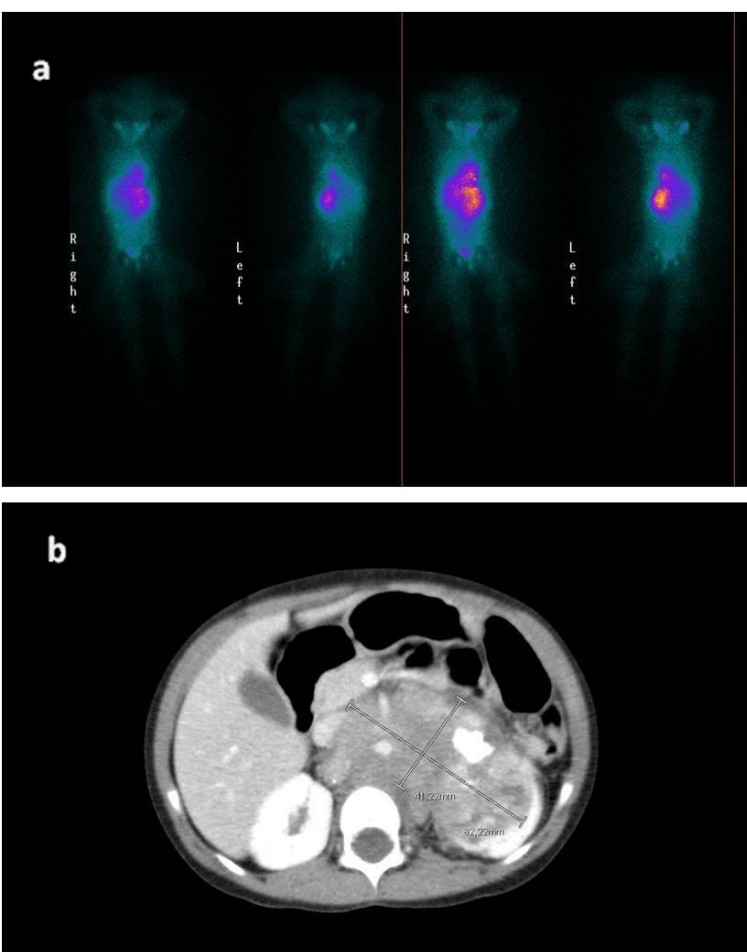

**Figure 1.** (**a**) Meta-(radioiodinated)iodobenzylguanidine showing abdominal mass uptake. (**b**) CT scan showing neuroblastoma.

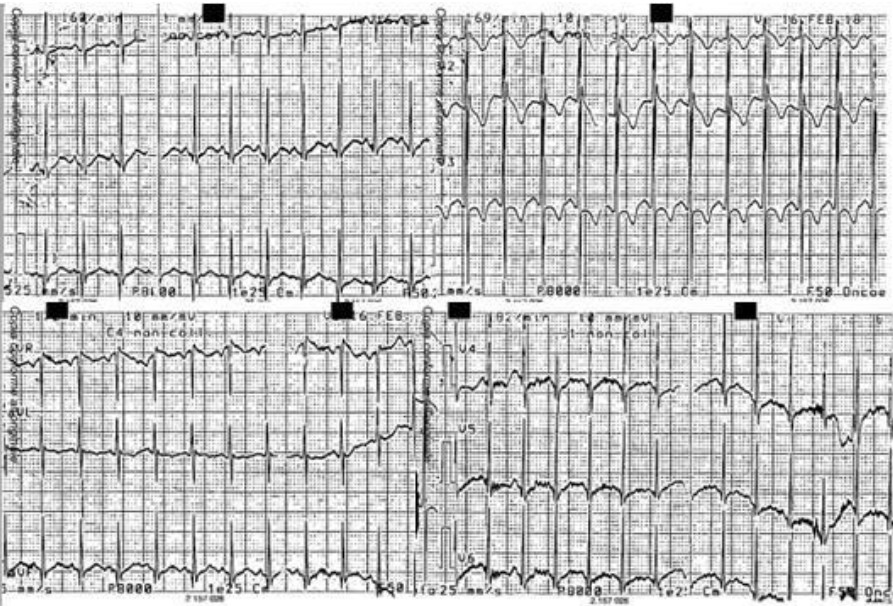

**Figure 2.** Electrocardiogram after cardiac resuscitation.

An exercise testing displayed that the mother had frequent premature ventricular beats during effort, disappearing at rest. Although the reported *RYR2* mutation was considered a variant of uncertain significance (VUS) after bioinformatics' analysis, the particular familial segregation may support its plausible pathogenicity. Cancer restaging after cardiac arrest, by means of (MIBG) single photon emission computed tomography (SPECT, GE Healthcare, Boston, MA, USA) and computer tomography (CT) scan, showed an important mass reduction and minimal uptake, while brain CT showed a picture of widespread suffering, but the clinical and neurological conditions of the child were good. On the basis of the cardiogenetic results, nadolol treatment was started, according to the current indications [11]; because of poor prognosis and low body mass index, cardiac defibrillator implantation was temporary avoided, with a strong indication to continue ECG monitoring during the treatment. A brief period of maintenance therapy with temozolomide and topotecan was useless, burdened by disease progression.

After multidisciplinary discussion, the need for treatment continuation was evident, avoiding the acknowledged cardiotoxicity; for this reason, the patient was administered a stepwise treatment by MIBG therapy, autologous peripheral blood stem cells (PBSC) infusion, followed by conditioning therapy with busulfan and melphalan and a second autologous PBSC infusion. Afterwards, she receiveda treatment with retinoic acid and effective immunotherapy based on Anti-GD2, a surface glycolipid, associated to improved survival and quality of life by reducing exposure to cytotoxic agents [12]. After twenty-eight months the patient is alive, without any evidence of further ventricular arrhythmias; unfortunately, cancer restaging by MIBG confirmed a tracer capturing mass. Currently, she is on etoposide maintenance treatment and continues cardiological follow up, according to the international consensus.

## 3. Methods

### 3.1. DNA Extraction, Libraries Preparation and Next-Generation Sequencing (NGS)

The patient's parents provided written informed consent to molecular testing and to the full content of this publication. This study was performed in accordance with the Declaration of Helsinki (1984) and its subsequent revisions. Genomic DNA was extracted from peripheral blood samples using the Bio Robot EZ1 (Quiagen, Solna, Sweden). DNA quantity and quality were measured by NanoDrop 2000 C Spectrophotometer (Thermo Fisher Scientific, Waltham, MA, USA). The sample was then quantified with the Qubit fluorometer (Thermo Fisher Scientific, Waltham, MA, USA) using the Quant-iT dsDNA BR Assay kit (Thermo Fisher Scientific, Waltham, MA, USA), according to the manufacturer's instructions. The sequencing of 75 genes responsible for several forms of cardiomyopathies and of 85 genes responsible for Hearing loss of the index patient was performed by using the SureSelectQXT Target Enrichment system (Agilent Technologies, Santa Clara, CA, USA), according to the manufacturer's instructions. The libraries were then sequenced on the NextSeq500 Sequencing System (Illumina Inc., San Diego, CA, USA), using a Mid Output 300 cycles flow cell (Illumina Inc., San Diego, CA, USA).

### 3.2. Sanger Sequencing

All putatively pathogenic variants were confirmed by Sanger sequencing and the segregation analysis was carried out. Sanger sequencing was performed using theBigDye Terminator v3.1 Cycle Sequencing kit (Applied Biosystems, Foster City, CA, USA) on the 3130xl Genetic Analyzer Sequencer (Thermo Fisher Scientific, Waltham, MA, USA).

### 3.3. Bioinformatics Analyses (vedi ref SGS)

The quality of the sequenced sequences was checked using the FastQC software [13] and then aligned against the hg19 human reference genome by the BWAmem v.0.7.17 [14]. Depth of coverage analysis was carried out through the TEQC R Package on the produced *.bam file, thus obtaining measures of sample and region-specific sequencing coverage [15]. The identification of single nucleotide variants and small insertions/deletions was per-

formed using GATK ver. 3.7 [16]. Variants were functionally annotated through the Annovar Software v.2017Feb15 [17], with the NCBI Human RefSeq as annotation reference system [18]. Annotated variants were also checked for novelty in public collections, such as dbSNP ver. 151 [19], ExAC and gnomAD [20]. Furthermore, prediction of functional effect of nonsynonymous and splice site variants were retrieved by the dbNSFP v3.4 database [21]. Subsequently, the prioritization of the variants started excluding those described as benign and likely benign. Then the remaining variants which passed this filtering were classified on the basis of their clinical relevance as pathogenic, likely pathogenic or variant of uncertain significance by using the following criteria: (i) nonsense/frameshift variant in genes previously described as disease causing by haploinsufficiency or loss-of-function; (ii) missense variant located in a critical or functional domain; (iii) variant affecting canonical splicing sites (i.e., ±1 or ±2 positions); (iv) variant absent in allele frequency population databases; (v) variant reported in allele frequency population databases but with a minor allele frequency (MAF) significantly lower than expected for the disease; (vi) variant predicted and/or annotated as pathogenic/deleterious in ClinVar v.2018Oct12 and/or LOVD v.2.0. The putative pathogenic variants identified following this pipeline were confirmed by direct Sanger sequencing, and the segregation analysis was carried out. The Sanger sequencing was performed using theBigDye Terminator v3.1 Cycle Sequencing kit (Applied Biosystems, Foster City, CA, USA) on the 3130xl Genetic Analyzer Sequencer (Applied Biosystems, Foster City, CA, USA). The clinical significance of the identified putative variants was interpreted according to the American College of Medical Genetics and Genomics Recommendations [22].

## 4. Results

The NGS analysis of the proband revealed that the patient was heterozygous for the *RYR2* (NM_001035.3):c.1927T > A (p.Leu643Ile) (rs368918887) variant, and homozygous for the *GJB2* (NM_004004) c.35delG, p.(Gly12Valfs*12) (rs80338939) variant. The detected variants were identified with a read depth of 604X and 1158X, respectively. After Sanger sequencing of the proband's parents, the *RYR2* variants resulted to be inherited from the mother, while both parents were heterozygous the *GJB2* variant.

## 5. Discussion

The occurrence of a major adverse cardiac event (MACE) gives rise to many questions in the clinical practice. The cardio-oncology approach requires the assessment of the patient's status and risk stratification before treatment for adequate risk assessment, patient'streatment and an appropriate follow up timing with close surveillance for side effects. In this case, preliminary cardiological assessment was normal, in particular cardiac contractility and ventricular repolarization on echocardiography and electrocardiogram. Serum electrolytes concentration was normal in the morning, before the cardiac arrest; in particular magnesium level, as it is an important RyR2 regulator, a potential mechanism by which Cisplatin may promote RyR2 hiperactivity duringhipomagnesemia. After cardiac arrest due to ventricular fibrillation, in the presence of long QT and congenital sensorineural deafness, the first hypothesis was represented by the Jarvell and Lange-Nielsen syndrome, characterized by an autosomal recessive inheritance pattern of potassium channel mutation, yet it seems questionable based on familial absence of deafness and sudden death. A first clinical approach based on epidemiological and probabilistic considerations can be helpful, leading to an initial intuitive orientation. However, in this case, the Ockham's razor showed its fallacy [23].

The NGS panel for genes responsible for hearing loss showed homozygosity for a pathogenic *GJB2* gene variant; previous studies denied a major incidence of cardiac arrhythmias and electrocardiographic parameter abnormalities (atrioventricular conduction or ventricular repolarization) in patients affected by this condition [24].

On the other hand, the presence of a VUS in the *RYR2* gene, with autosomal dominant inheritance pattern, even in the presence of a weak familial segregation, represented by

maternal history of palpitation, syncope and frequent premature ventricular beats during effort, brings a reflection on the strict role of intracellular calcium handling with implication on ventricular fibrillation induction.

The *RYR2* gene is implicated in the pathogenesis of catecholaminergic polymorphic ventricular tachycardia (CPVT), a rare inherited arrhythmia syndrome characterized by adrenergically driven ventricular arrhythmia predominantly caused by pathogenic variants in the cardiac ryanodine receptor (RyR2) [25]. RyR2 is responsible for sarcoplasmic reticulum calcium release, hence its dysfunction leads to an impairment of intracellular calcium handling [26]. On the other hand, although a direct interaction between RyR2 and cisplatin cannot be ruled out [27], it is described that cisplatin induces an irreversible inhibition of SERCa2, reducing calcium reuptake in sarcoplasmic reticulum [28]. Broadening the comprehension of the complex biological system interactions during chemotherapy, we need to consider patient psychological distress, leading to anxiety and increased adrenergic burden, well described by the recent paradigm of "emotional chemobrain" [29]. As described by Trafford et al., beta adrenergic stimulation induces calcium overload mainly by RyR2 phosporylation, increasing $Ca^{2+}$ leaking and enhancing L-type $Ca^{2+}$ current, with consequent increase in the intracellular calcium concentration. High $Ca^{2+}$ concentration leads to both $Ca^{2+}$-induced $Ca^{2+}$ release phenomenon (CICR) by SR and Na-Ca pump exchange (NCX) activation, causing $Na^+$ inward current and delayed after-depolarization (DAD) [30]. Summarizing, patient concealed predisposition to cardiac arrhythmias, represented by pathological calcium release due to the *RYR2* mutation, is worsened by further calcium intracellular concentration, secondary to reduced calcium uptake by cisplatin-induced SERCa2 inhibition and stress-related beta-stimulation, impairs cytosolic $Ca^{2+}$ waves propagation and full development [31]; this complex processleads to early and delayed after depolarizations (EADs and DADs), precipitating in ventricular fibrillation [32].

According to the international cardiopulmonary resuscitation protocol [8], epinephrine is strongly indicated. In our case, although a prompt intervention was guaranteed, ROSC was obtained after one hour. Our plausible consideration is that epinephrine has a detrimental effect in the presence of the *RYR2* mutation, leading to further calcium overload. The recent literature supports our hypothesis, confirming that epinephrine administration during cardiac arrest in children affected by CPVT is associated to significantly longer resuscitation efforts, more interventions and a longer time to return to spontaneous circulation [33]. Starting from this initial evidence, strengthened by biological plausibility, we suggest that a different protocol needs to be discussed and validated in pediatric patients affected by CPVT. In particular, vasopressine, recently ruled out in the setting of cardiac arrest by the 2017 AHA/ACC/HRS Guideline [34], with a current renewed adoption in pediatric intensive care [35], could be a valid alternative in this particular subgroup and represents an interesting hypothesis to investigate in further studies.

After multidisciplinary discussion, even in the presence of an indication to ICD implantation in secondary prevention after the cardiac arrest, we decided not to implant for several reasons mainly represented by low body mass index with higher risk of complication [36], poor life expectancy and the presence of a reversible cause of ventricular fibrillation represented by cisplatin infusion. On the other hand, the evidence of disease progression needs to be addressed through a different available treatment, in particular by immunotherapy, dinutuximab, an anti-GD2 chimeric monoclonal antibody approved for high-risk neuroblastoma treatment [37]. In our case, according to the literature, this solution avoids the occurrence of any further arrhythmias, which is made evident by continuous ECG monitoring during hospitalization.

Currently, pharmacogenomic focuses mostly on pharmacokinetics, in particular on drug metabolism modification induced by gene polymorphism, implicating different drug-response and side effect distribution among population; this approach is suffering from some limitations, due to cost-effectiveness and lack of reliable predictable models [38]. Future aims of pharmacogenomics will rely on a better characterization of the patient's

whole genome to build predictable models and reach a personalized medicine by using big data analysis platforms [39].

## 6. Conclusions

In conclusion, existing evidence demonstrates that genetic factors have the potential to improve the discrimination between individuals at higher and lower risk of cardiac complications, such as heart failure and life-threatening cardiac arrhythmias. We underline the role of genetic analysis on this issue, allowing the best support to patient care and healthcare decision making, improving prevention of cardiotoxicity. In the future, this combined approach will allow a correct recognition of this demanding characterization, bringing together oncological and cardiological features of this complex scenario, leading to proper clinical management in both the acute and the chronic patient pharmacological treatment.

**Author Contributions:** Data acquisition: A.M., S.M., G.D.S. and M.V.; processing and interpretation, A.M., S.M., G.D.S. and D.R.P.; writing original draft preparation, A.M., S.M., G.D.S., M.V. and M.P.S.; clinical evaluation, A.M., S.M., G.D.S., A.S., M.V. and M.P.S.; study concept, S.M., G.D.S., S.L., D.R.P. and M.V.; critical revision of manuscript, A.M., S.M., G.D.S. and M.C. (Massimo Carella); data curation, S.M., G.D.S., P.P., S.C. and M.P.L.; supervision, G.D.S., D.R.P., M.C. (Marco Castori), M.V., M.P.S. and M.C. (Massimo Carella). All authors have read and agreed to the published version of the manuscript.

**Funding:** This work was supported by Italian Ministry of Health for research grant RC 2001CA08.

**Institutional Review Board Statement:** Not applicable.

**Informed Consent Statement:** Informed consent, approved by Fondazione IRCCS Casa Sollievodella-Sofferenza Ethical Committee, was obtained by both parents for the genetic analysis and case history.

**Data Availability Statement:** The datasets used and/or analyzed during the current study are available from the corresponding author on reasonable request.

**Conflicts of Interest:** The authors declare no conflict of interest.

## Abbreviations

HR-NB: High Risk Neuroblastoma

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
