# Peer review of "Pharmacogenomics of Pediatric Cardiac Arrest: Cisplatin Treatment Worsened by a Ryanodine Receptor 2 Gene Mutation"

_cardiogenetics, doi:10.3390/cardiogenetics12010007_

Round 1

Reviewer 1 Report

The case report is very interesting and well described. Some comments try to introduce constructive suggestions in case the authors decide to follow them. 

-The title is too long in my opinion, just as a suggestion I would keep it as " Pharmacogenomic of Pediatric Cardiac Arrest: Cisplatin Treatment Worsened by a Ryanodine Receptor 2 Gene Mutation"

  • Given that Magnesium is an important RyR2 regulator, it would be important to report Magnesemia in the patient. A potential mechanism in which Cisplatin promotes RyR2 hiperactivity due to hipomagnesemia should be considered.
  • On the other hand, a possible direct interaction between cisplatin and RyR2 should be considered if possible due to bioinformatics. Previos reports demonstrating direct interaction between RyR2 and antitumoral compounds would be a good reference to be introduced (Chakraborty AD, Gonano LA, Munro ML, Smith LJ, Thekkedam C, Staudacher V, Gamble AB, Macquaide N, Dulhunty AF, Jones PP. Activation of RyR2 by class I kinase inhibitors. Br J Pharmacol. 2019 Mar;176(6):773-786. doi: 10.1111/bph.14562. Epub 2019 Jan 30. PMID: 30588601; PMCID: PMC6393232.)
  • The refered action of cisplatin on SERCA seems to be opposite to the development of arrhythmias related with SR overload and RyR2 spontenaous opening. One possible reconciliation could be that a partial inhibition of reuptake affect waves propagation and full development (Valverde CA, Mazzocchi G, Di Carlo MN, Ciocci Pardo A, Salas N, Ragone MI, Felice JI, Cely-Ortiz A, Consolini AE, Portiansky E, Mosca S, Kranias EG, Wehrens XHT, Mattiazzi A. Ablation of phospholamban rescues reperfusion arrhythmias but exacerbates myocardium infarction in hearts with Ca2+/calmodulin kinase II constitutive phosphorylation of ryanodine receptors. Cardiovasc Res. 2019 Mar 1;115(3):556-569. doi: 10.1093/cvr/cvy213. PMID: 30169578; PMCID: PMC6383052.)
  • The most important change I suggest is to explain better the mechanism of RyR2 mediated leak-arrhythmias, which is not promoted straightforward by impairment of SERCA function, because it depends of SR calcium load, which will be reduced by lower uptake. You will find information about this mechanism by looking at the explanation of Store-overload induced Ca2+ release (SOICR) or in this review written by Eisner and Trafford (https://pubmed.ncbi.nlm.nih.gov/19667488/)

Author Response

  Dear Reviewer,

thank to support our manuscript by your suggestions.

  • According to your indications we modify the title as you can read.
  • We agree with your concern about magnesium levels; we check the patient files, finding normal concentration . We specify this concept in the text.
  • We agree on your suggestion,  as this allow us to improve our manuscript quality, as you can read in the next revisioned version. In particolar:
  • We cite the possible direct mechanism interaction between RyR2 and cisplatin, by short mentioning in the manuscript , referring to Br J Pharmacol. 2019 Mar;176(6):773-786.
  • We describe a possible mechanism of SR calcium overload mediated by beta-adrenergic stimulation secondary to psychological distress, introducing the concept of Emotional Chemobrain.
  • Following your suggestion ,we further explain the hypotesis of calcium-related induced arrhythmias citing Trafford.

Reviewer 2 Report

Thank you for allowing me to review the following manuscript titled "Pharmacogenomic of Calcium Handling in Pediatric Cardiac 2
Arrest: Cisplatin Treatment Worsened by a Ryanodine Recep- 3
tor 2 Gene Mutation" by Angela et al.

Oversll well written but I have some revisions on the case report

  1. Change the title to "pharmacogenomics"
  2. It is better to elaborate what constituted "Preliminary cardiological screening was normal. "

Author Response

Dear Reviewer,

we thank you for improving our manuscript by your suggestions.

According to your indication,

  • we change the title
  • and clarify preliminary cardiological screening in the text.
